# Pyridinone Derivatives as Interesting Formyl Peptide Receptor (FPR) Agonists for the Treatment of Rheumatoid Arthritis

**DOI:** 10.3390/molecules26216583

**Published:** 2021-10-30

**Authors:** Letizia Crocetti, Claudia Vergelli, Gabriella Guerrini, Maria Paola Giovannoni, Liliya N. Kirpotina, Andrei I. Khlebnikov, Carla Ghelardini, Lorenzo Di Cesare Mannelli, Elena Lucarini, Igor A. Schepetkin, Mark T. Quinn

**Affiliations:** 1NEUROFARBA, Pharmaceutical and Nutraceutical Section, University of Florence, Via Ugo Schiff 6, 50019 Sesto Fiorentino, Italy; letizia.crocetti@unifi.it (L.C.); claudia.vergelli@unifi.it (C.V.); gabriella.guerrini@unifi.it (G.G.); 2Department of Microbiology and Cell Biology, Montana State University, Bozeman, MT 59717, USA; liliya.kirpotina@montana.edu (L.N.K.); igor@montana.edu (I.A.S.); mquinn@montana.edu (M.T.Q.); 3Kizhner Research Center, Tomsk Polytechnic University, 634050 Tomsk, Russia; aikhl@chem.org.ru; 4Scientific Research Institute of Biological Medicine, Altai State University, 656049 Barnaul, Russia; 5NEUROFARBA, Pharmacology and Toxicology Section, University of Florence, Viale Pieraccini 6, 50139 Florence, Italy; carla.ghelardini@unifi.it (C.G.); lorenzo.mannelli@unifi.it (L.D.C.M.); elena.lucarini@unifi.it (E.L.)

**Keywords:** pyridinone, formyl peptide receptors, agonists, rheumatoid arthritis, molecular docking

## Abstract

Rheumatoid arthritis (RA) is a chronic inflammatory disease characterized by joint inflammation, cartilage damage and bone destruction. Although the pharmacological treatment of RA has evolved over the last few years, the new drugs have serious side effects and are very expensive. Thus, the research has been directed in recent years towards new possible targets. Among these targets, N-formyl peptide receptors (FPRs) are of particular interest. Recently, the mixed FPR1/FPR2 agonist **Cpd43**, the FPR2 agonist **AT-01-KG**, and the pyridine derivative **AMC3** have been shown to be effective in RA animal models. As an extension of this research, we report here a new series of pyridinone derivatives containing the (substituted)phenyl acetamide chain, which was found to be essential for activity, but with different substitutions at position 5 of the scaffold. The biological results were also supported by molecular modeling studies and additional pharmacological tests on **AMC3** have been performed in a rat model of RA, by repeating the treatments of the animals with 10 mg/kg/day of compound by 1 week.

## 1. Introduction

Rheumatoid arthritis (RA) is a chronic inflammatory autoimmune disease characterized by autoantibody production, joint inflammation, synovial hyperplasia, cartilage damage, and bone destruction, as well as cardiovascular, hematological, and pulmonary complications [1]. RA affects about 0.5–1% of the population in the world and is characterized by progressive disability and increased mortality, reducing life expectancy on average by 5–10 years [2]. RA is treated with glucocorticoids and conventional synthetic disease-modifying antirheumatic drugs (csDMARDs), such as methotrexate. More recently, the powerful and selective biologic DMARDs (bDMARDs) and targeted synthetic DMARDs (tsDMARDs) have been added to the pharmaceutical armamentarium, but these drugs have serious side effects and are very expensive [3]. Moreover, these therapeutic approaches fail in a substantial number of patients; hence, there is significant interest in alternative targets for RA treatment [4].

Among the numerous and different cells types, cytokines and endogenous mediators involved in the pathogenesis of RA, special attention has been paid to annexin A1 (AnxA1) [5], an anti-inflammatory mediator able to inhibit leukocyte recruitment, pro-inflammatory cytokine expression, and joint inflammation, as demonstrated in several RA animal models [5,6]. Recently, it has been demonstrated that AnxA1 exerts part of its anti-inflammatory effects by interacting with N-formyl peptide receptors (FPRs), especially FPR2 and to a lesser extent FPR1 [5,6]. AnxA1 and FPR2 are expressed by fibroblast-like synoviocytes (FLS), which play a critical role in the pathogenesis of RA by inducing the production of proteases responsible for degradation of the extracellular matrix as well as the production of cytokines by immune cells [6]. It has been demonstrated that AnxA1 suppresses FLS production of pro-inflammatory cytokines, acting as an agonist of FPR2 [5,6,7].

FPR2 and FPR1, together with FPR3, belong to the family of G protein-coupled receptors and although they share a significant sequence homology [8], they have different functional properties and are differently expressed in the various cells. FPRs are present in many immune cells, including neutrophils (except FPR3), T lymphocytes, monocytes, and macrophages, as well as in non-immune cells, such as astrocytes, hepatocytes, and endothelial cells. FPR1 is also expressed on myeloid cells, while FPR2 is found in B cells, as well as in activated/memory T cells [9,10]. FPR2 binds various endogenous ligands, including AnxA1, lipoxin A4 (LXA4) and serum amyloid A (SAA), whereas FPR1, in addition to AnxA1, also binds cathepsin G, formylated bacteria, and mitochondrial peptides. Interestingly, FPR2 can have an antiinflammatory effect depending on the nature of the ligand. The binding with anti-inflammatory mediators AnxA1 or LXA4 results in FPR2 receptor homodimerization and in the release of inflammation-resolving cytokines and neutrophil apoptosis. Conversely, binding of inflammatory ligands, such as serum amyloid A (SAA), does not produce FPR2 homodimerization, resulting in increased expression of pro-inflammatory cytokines [11,12].

The important role of FPRs in inflammation makes these receptors, in particular FPR2, potential targets for the treatment of some inflammatory diseases, such as RA [13]. For example, the FPR1/FPR2 mixed agonist Cpd43 (Figure 1) synthesized by Amgen, proved to be beneficial in animal models (collagen-induced and antigen-induced arthritis) of RA [14], significantly reducing the tissue inflammation, cartilage damage, and synovitis [6] at a dose of 30 mg/kg (i.p.). Pre-treatment of the animals with FPR antagonists (Boc-MLF and WRW4) resulted in complete reversal of the effects of Cpd43, thus demonstrating the direct correlation between FPR activation by Cpd43 and reduction of arthritis severity. Likewise, the FPR2 agonist AT-01-KG (Figure 1) has been shown to exhibit anti-inflammatory activity in a model of antigen-induced arthritis [15,16] by increasing neutrophil apoptosis and subsequent efficient efferocytosis.

Over the years, we reported numerous compounds with various scaffolds and different potency/selectivity towards the three FPR isoforms [17,18,19,20,21,22,23,24,25]. Our latest research involved the study of pyridinone derivatives, which showed some selectivity towards FPR2 [24]. The most interesting compound **AMC3** (**2a** in the original paper) (Figure 2) was evaluated in vivo in a rat model of rheumatoid arthritis and was able to completely control pain hypersensitivity at a dose of 30 mg/kg/day. A common requirement for all previously published active compounds was the presence of the substituted phenyl acetamide fragment (bromine was the best substituent) and a methoxyphenyl ring linked to the heterocyclic scaffold. Surprisingly, when the methoxyphenyl ring at position 5 in the series of pyridinone derivatives was replaced with an ethoxy carbonyl group as in compound **AMC4** (**2d** in the original paper [24]) (Figure 2), the activity and FPR2 selectivity were preserved.

Based on these results, we directed our research toward the synthesis of additional pyridinone derivatives that retained the phenylacetamide chain in each case but were differently substituted at position 5 of the pyridinone scaffold. In particular, we synthesized compounds lacking the methoxyphenyl group (5-ester and 5-amide alkyl derivatives) or compounds that still had the methoxyphenyl group but were spaced from the pyridinone by a ketone or an ester group. Finally, some products containing one or two OH groups on the phenylacetamide chain and/or on the phenyl at position 5 of the scaffold were synthesized in order to understand whether the inclusion of this group affected the activity profile (Figure 2).

## 2. Results and Discussion

### 2.1. Chemistry

All new compounds were synthetized following the procedures described in Figure 1, Figure 2, Figure 3 and Figure 4, and the structures were unequivocally confirmed on the basis of analytic and spectral data. Figure 1 shows the synthetic pathway used to obtain the final compounds **2** and of type **4**, acid and ester derivatives, respectively. Starting from the key intermediate **1**, synthesized following procedures reported in the literature [26], final compound **2** was obtained by direct alkylation with 4-bromophenyl-2-chloroacetamide [27] in anhydrous CH_3_CN and K_2_CO_3_.

For synthesis of the ester derivatives **4a–e**, compound **1 [26]** was first treated with SOCl_2_, then with the appropriate aryl or alkyl alcohol and a catalytic amount of Et_3_N in THF, resulting in intermediates **3a–e** (**3b** and **3c** [28]). The intermediates were alkylated with a 4-bromophenyl-2-chloroacetamide [27] chain under the conditions above reported for compound **2**.

The synthetic procedure shown in Figure 2 summarizes synthesis of the 3-carboxamide derivatives **6** and of type **8**. Intermediate **5** was obtained by treatment of the same key precursor **1** [26] with 1,1′-carbonyldiimidazole (CDI) and 28% NH_4_OH in anhydrous THF, and then, it was alkylated with 4-bromophenyl-2-chloroacetamide [27] under the conditions previously described, resulting in the final compound **6**. The carboxamides of type **8** were obtained following the same procedure described in Figure 1 through the acyl chloride, which was treated with the suitable alkylamine, resulting in intermediates **7a–e**, which were finally transformed into **8a–e** by direct alkylation.

Figure 3 shows the synthesis of compounds **10** and **11**, which contain a carbonyl function as spacer between the pyridinone scaffold and the methoxybenzoyl group. Treatment of precursor **1 [26]** with Eaton’s reagent and anisole, in addition to the insertion of the p-methoxybenzoyl group at position 5, resulted in hydrolysis of the CN group (intermediate **9**). This was alkylated with 4-bromophenyl-2-chloroacetamide chain [27], resulting in compound **10,** which was further subjected to dehydration with POCl_3_ to obtain the final 3-ciano derivative **11**.

Finally, Figure 4 shows the synthesis of final compounds **13a–c** (**13a** is the previously reported **AMC3 [24]**) and **14a–c**, which have one or two OH groups on the phenyl rings of the fragment at N-1 and/or on the phenyl at position 5 of the scaffold. Pyridinones **13a–c** were obtained following the same procedure reported in the previous schemes by using the appropriate (4/5-substituted phenyl)-2-chloroacetamides [27,29,30]. Treatment of compounds of type **13** with a solution of 1M BBr_3_ in CH_2_Cl_2_ at room temperature resulted in the final 1**4a–c.**

### 2.2. Biological Results

All compounds were tested for their ability to induce intracellular Ca^2+^ flux in human neutrophils (hPMN) and in human HL-60 cells transfected with either FPR1 or FPR2. The results are reported as EC_50_ values in Table 1 and Table 2 using as reference compounds *f*MLF (FPR1 agonist), WKYMVM (FPR2 agonist), and our previously described agonists **AMC3** and **AMC4** [24]. All new compounds were also evaluated in non-transfected HL-60 wild-type cells and were found to be inactive.

The biological results reported in Table 1 and Table 2 demonstrate that most compounds acted as FPR1 and FPR2 agonists in the micromolar range and had lower selectivity for FPR2 than the lead compound **AMC3**. Moreover, they exhibited a comparable activity on hPMN. We also found that position 5 of the scaffold can be extensively modified, although the derivatives were less potent than **AMC3** and **AMC4**. In fact, the insertion of alkyl esters (compounds **4a**–**c**), the maintenance of the methoxyphenyl group variously spaced from the pyridinone scaffold with ester or carbonyl groups (compounds **4d, 4e**, **10,** and **11**), as well as the presence of a hydroxy carbonyl group (compound **2**) all resulted in retention of FPR1/FPR2 mixed agonist activity (Table 1). Similar results can be observed in Table 2 for amide derivatives **6** and of type **8**, which exhibited EC_50_ values in the low micromolar range and a moderate FPR2 selectivity for compounds **8d** and **8e** that was similar to **AMC4** selectivity index (SI) FPR1/FPR2 = 9 and 7.5, respectively.

Comparing the 5-carbethoxy derivative **AMC4** and its isostere **8b**, it can be seen that replacement of the ester group by -CONH- is responsible for the decrease in activity, despite maintaining a small selectivity for FPR2 (SI = 8 and 6 for **AMC4** and **8b,** respectively). Finally, the introduction of one or two hydroxylic groups on the phenyl ring of the side chain or at position 5 of the pyridinone scaffold resulted in inactive (**13b,c** and **14c**) or very low activity (**14b**) compounds, with the exception of **14a**, the nor-derivate of **AMC3**, which had a mixed FPR1/FPR2 agonist profile with submicromolar EC_50_ values (EC_50_ = 0.23 μM for FPR1 and FPR2) (Table 2).

In previous studies, we analyzed one dose of **AMC3** in a rat model of RA. Here, we further evaluated **AMC3** [24] in this model using repeated treatments with 10 mg/kg/day of **AMC3** for 1 week. The rats were treated once intraarticularly (left paw, ipsilateral) with complete Freund’s adjuvant (CFA) on day 1, and **AMC3** (10 mg/kg) was administered p.o. daily starting on the same day. Behavioral tests were performed on day 8, 24 h after the last administration of **AMC3**. CFA-treated rats had a reduced pain threshold to the noxious mechanical stimulus (paw pressure test) in comparison with sham + vehicle animals (35.2 ± 1.6 g for CFA + vehicle vs. 66.4 ± 2.3 g for sham + vehicle, whereas AMC3 significantly increased the pain threshold to 52.4 ± 2.8 g (Figure 3, Panel A). 

CFA-induced hypersensitivity was also evaluated by an incapacitance test, where the change in hind limb weight bearing was measured as the difference (Δ Weight) between the weight burden on the contralateral (vehicle-treated) and the ipsilateral (CFA-treated) limbs. On day 8, CFA-treated rats had an increased postural unbalance in comparison with sham + vehicle animals (57.2 ± 4.0 g for CFA + vehicle vs 3.6 ± 2.5 g for sham + vehicle), while **AMC3** significantly reduced the postural unbalance (34.6 ± 3.8 g for CFA + **AMC3**) (Figure 3, Panel B). Therefore, compound **AMC3** was active against rheumatoid arthritis RA-related hypersensitivity when repeatedly administered. These findings are in agreement with the previously published results [24] regarding the acute pain-relieving activity of **AMC3** and suggest the therapeutical potential of **AMC3** when used for repeated treatment.

### 2.3. Molecular Docking

In order to understand differences between active compounds **AMC4**, **14a** and inactive compound **14c** in FPR1 and FPR2 binding, we performed molecular docking analysis of these molecules. For this purpose, we used a homology model of FPR1 and the experimentally obtained FPR2 structure (PDB entry 6OMM) published recently by Zhuang et al. [31]. We also analyzed the corresponding docking poses of active compound **AMC3**, which we reported previously [25]. The docking results obtained with FPR1 showed that the investigated compounds are anchored in the binding site by H-bonding with Tyr257 and some other residues (Table 3). The docking poses are presented in Appendix A. Recently [25], we determined that **AMC3** in its docking pose interacts with FPR1 by H-bonding to Arg201, Gln258, and Ser287. In spite of the different H-bonding patterns, the three active compounds **AMC3**, **AMC4**, and **14a** had very similar positionings of their molecular fragments within the binding site. The overlaid *p*-bromophenyl groups attached to the acetamide moiety (Figure 4, Panel A) are pulled into a “hole” directed from the binding site to the outside of the receptor [25].

The pyridinone carbonyl group of compound **14c** is H-bonded to Tyr257 analogously to **AMC4** and **14a** (Table 3, Appendix A). However, the N-phenylacetamido and phenylpyridinone moieties of **14c** are interchanged in space with respect to the corresponding moieties of the active compounds **AMC3**, **AMC4**, and **14a** (see, for example, the superimposed docking poses of ligands **AMC3** and **14c** in Figure 4, Panel B). Both hydroxy groups of **14c** form hydrogen bonds with Arg201 and Arg205 (Table 3, Appendix A), leading to a different binding mode in the receptor site.

Analogous results were obtained on the docking of compounds **AMC4**, **14a**, and **14c** in the FPR2 binding site (see the poses in Appendix A). The residues participating in H-bonding with the ligands are given in Table 3. Previously, we found that compound **AMC3** has a docking pose in FPR2 with H-bonding to Asp106 [25]. Although the H-bonding patterns of the active compounds **AMC3**, **AMC4**, and **14a** were quite different, they are located similarly within the FPR2 binding site (see Figure 5, Panel A showing the superimposed poses). The overlaid *p*-bromophenyl moieties again occupy a “hole” directed from the FPR2 cavity to the outside of the receptor.

In contrast, the inactive compound **14c** is anchored by hydrogen bonds to Arg201 and Gln258 in a position that is reversed with respect to the active compounds **AMC3**, **AMC4**, and **14a**. Thus, the substituted phenylpyridinone fragment of **14c** almost coincides with the *p*-bromophenyl acetamido moiety of ligand **AMC3** (Figure 5, Panel B). In this case, the *p*-hydroxyphenyl group of **14c** does not overlay with the rest of the **AMC3** molecule.

Overall, our docking results support the earlier observation that the presence of a bromine atom in the *para*-position of phenyl acetamido group is important for FPR1/FPR2 activity of this series of pyridinone compounds. These results agree with the previously obtained data indicating that the *p*-bromophenyl group of active *p*-bromophenyl acetamides effectively enters the “hole” area of FPR1 and FPR2 [25]. Replacing the bromine atom with a hydroxy group drastically changes the ligand binding mode. Conversely, position 5 of the pyridinone scaffold seems to tolerate different types of substituents without producing important variation in interactions with the receptor binding site.

## 3. Materials and Methods

### 3.1. Physical Measurements and Materials

Reagents and starting materials were obtained from commercial sources. Extracts were dried over Na_2_SO_4_, and the solvents were removed under reduced pressure. Thin layer chromatography (TLC) was used to check the progress of reactions (plates were precoated with Merck silica gel 60 F-254) and the visualization was performed through UV fluorescence (λ_max_ = 254 nm). The techniques used to separate the mixtures are gravity chromatography (Kieselgel 40, 0.063–0.200 mm; Merck) and flash chromatography (Kieselgel 40, 0.040–0.063 mm; Merck) and yields were calculated on chromatographically and spectroscopically pure products. Compounds were named following IUPAC rules, as applied by Beilstein-Institut AutoNom 2000 (4.01.305) or CA Index Name. All melting points were determined on a microscope hot stage Büchi apparatus and are uncorrected. The identity of all intermediates and final compounds was ascertained through ^1^HNMR and ^13^C NMR. Spectra which were recorded on an Avance 400 instruments (Bruker Biospin Version 002 with SGU). Chemical shifts (*δ*) are reported in ppm to the nearest 0.01 ppm using solvent as the internal standard. Coupling constants (*J* values) are given in Hz and were calculated using ‘TopSpin 1.3′ software Nicolet Instrument Corp., Madison, WI) and rounded to the nearest 0.1 Hz. Compound purity: microanalyses indicated by the symbols of the elements or functions were performed with a Perkin-Elmer 260 elemental analyzer for C, H, and N, and they were within ± 0.4 % of the theoretical values.

### 3.2. General Procedure for the Synthesis of New Compounds

#### 3.2.1. General Procedure for **2** and **4a–e**

To a mixture of appropriate compounds **1** [26] or **3a–e** (0.27 mmol) (**3b** and **3c** were previously described [28]) and 0.54 mmol of K_2_CO_3_ in 2-5 mL of anhydrous CH_3_CN, 0.27–0.41 mmol of 4-bromophenyl-2-choroacetamide [27] was added, and the mixture was refluxed under stirring for 4–18 h (compound **2** for 30 h). After cooling, the suspension was concentrated in vacuo, 10 mL of ice-cold water was added, and the precipitate was recovered by suction. The final compounds were purified by flash chromatography using toluene/ethyl acetate/acetic acid 7:3:1 for compound **2**; dichloromethane/methanol 9.5:0.5 for compounds **4a,b** and **4d,e**; and cyclohexane/ethyl acetate 1:2 for **4c,** as eluents. All final compounds were crystallized from ethanol.

*1-{2-[(4-Bromophenyl)amino]-2-oxoethyl}-5-cyano-2-methyl-6-oxo-1,6-dihydropyridine-3-carboxylic acid* (**2**), Yield = 22%; mp = 180–182 °C dec. (EtOH); ^1^H-NMR (DMSO-d_6_) δ 2.80 (s, 3H, CH_3_); 4.98 (s, 2H, CH_2_); 7.50 (m, 4H, Ar); 8.48 (s, 1H, Ar); 10.70 (exch br s, 1H, NH). ^13^C-NMR (DMSO-d_6_) δ 18.29 (CH_3_); 48.48 (CH_2_); 98.65 (C); 115.67 (C); 116.11 (C); 116.71 (C); 121.65 (CH); 132.13 (CH); 138.41 (C); 149.48 (CH); 158.19 (C); 160.25 (C); 165.44 (C); 167.07 (C). Anal. Calcd for C_16_H_12_BrN_3_O_4_:C, 49.25; H, 3.10; N, 10.77; found: C, 49.38; H, 3.11; N, 10.75.

*Propyl 1-{2-[(4-bromophenyl)amino]-2-oxoethyl}-5-cyano-2-methyl-6-oxo-1,6-dihydropyridine-3-carboxylate* (**4a**)**,** Yield = 15%; mp = 193–195 °C dec. (EtOH); ^1^H-NMR (DMSO-d_6_) δ 0.93 (t, 3H, CH_2_*CH_3_*, *J* = 8.0 Hz); 1.69 (m, 2H, *CH_2_*CH_3_); 2.73 (s, 3H, CH_3_); 4.16 (t, 2H, OCH_2_, *J* = 8.0 Hz); 5.02 (s, 2H, N-CH_2_); 7.50 (m, 4H, Ar); 8.50 (s, 1H, Ar); 10.62 (exch br s, 1H, NH). ^13^C-NMR (DMSO-d_6_) δ 10.88 (CH_3_); 18.78 (CH_3_); 21.87 (CH_2_); 48.87 (CH_2_); 67.26 (CH_2_); 99.78 (C); 109.99 (C); 115.81 (C); 115.94 (C); 121.57 (CH); 132.00 (CH); 138.25 (C); 148.23 (CH); 159.88 (C); 160.81 (C); 164.48 (C); 165.01 (C). Anal. Calcd for C_19_H_18_BrN_3_O_4_: C, 52.79; H, 4.20; N, 9.72; found: C, 52.61; H, 4.21; N, 9.74.

*Butyl 1-{2-[(4-bromophenyl)amino]-2-oxoethyl}-5-cyano-2-methyl-6-oxo-1,6-dihydropyridine-3-carboxylate* (**4b**), Yield = 14%; mp = 109–111 °C dec. (EtOH); ^1^H-NMR (DMSO-d_6_) δ 0.90 (t, 3H, CH_2_*CH_3_*, *J* = 8.0 Hz); 1.37 (m, 2H, *CH_2_*CH_3_); 1.66 (m, 2H, CH_2_*CH_2_*), 2.72 (s, 3H, CH_3_); 4.20 (t, 2H, OCH_2_, *J* = 8.0 Hz); 5.02 (s, 2H, N-CH_2_); 7.50 (m, 4H, Ar); 8.50 (s, 1H, Ar); 10.62 (exch br s, 1H, NH). ^13^C-NMR (DMSO-d_6_) δ 14.05 (CH_3_); 18.77 (CH_3_); 19.18 (CH_2_); 30.51 (CH_2_); 48.87 (CH_2_); 65.49 (CH_2_); 99.78 (C); 109.98 (C); 115.80 (C); 115.94 (C); 121.58 (CH); 132.00 (CH); 138.26 (C); 148.22 (CH); 159.88 (C); 160.81 (C); 164.46 (C); 165.01 (C). Anal. Calcd for C_20_H_20_BrN_3_O_4_: C, 53.82; H, 4.52; N, 9.42; found: C, 53.97; H, 4.51; N, 9.47.

*Isopropyl 1-{2-[(4-bromophenyl)amino]-2-oxoethyl}-5-cyano-2-methyl-6-oxo-1,6-dihydropyridine-3-carboxylate* (**4c**), Yield = 19%; mp = 223–224 °C (EtOH); ^1^H-NMR (DMSO-d_6_) δ 1.29 (d, 6H, CH(*CH_3_*)*_2_*, *J* = 6.4 Hz); 2.72 (s, 3H, CH_3_); 5.01(s, 2H, N-CH_2_); 5.05 (m, 1H, *CH*(CH_3_)_2_); 7.50 (m, 4H, Ar); 8.49 (s, 1H, Ar); 10.61 (exch br s, 1H, NH). ^13^C-NMR (DMSO-d_6_) δ 18.73 (CH_3_); 21.96 (CH_3_); 48.83 (CH_2_); 69.53 (CH); 99.76 (C); 110.34 (C); 115.82 (C); 115.95 (C); 121.61 (CH); 132.18 (CH); 138.26 (C); 148.25 (CH); 159.88 (C); 160.62 (C); 163.92 (C); 165.02 (C). Anal. Calcd for C_19_H_18_BrN_3_O_4_: C, 52.79; H, 4.20; N, 9.72; found: C, 52.63; H, 4.20; N, 9.74.

*3-Methoxybenzyl 1-{2-[(4-bromophenyl)amino]-2-oxoethyl}-5-cyano-2-methyl-6-oxo-1,6-dihydropyridine-3-carboxylate* (**4d**), Yield = 20%; mp = 140–143 °C (EtOH); ^1^H-NMR (DMSO-d_6_) δ 2.73 (s, 3H, CH_3_); 3.74 (s, 3H, OCH_3_); 5.02 (s, 2H, N-CH_2_); 5.24 (s, 2H, OCH_2_); 6.90 (d, 1H, Ar, *J* = 8 Hz); 7.03 (s, 2H, Ar); 7.29 (t, 1H, Ar, *J* = 8 Hz); 7.50 (m, 4H, Ar); 8.54 (s, 1H, Ar); 10.62 (exch br s, 1H, NH). 13C-NMR (DMSO-d6) δ 18.86 (CH3); 48.91 (CH2); 55.57 (CH3); 67.16 (CH2); 99.81 (C); 109.67 (C); 114.11 (CH); 114.27 (CH); 115.91(C); 120.72 (CH); 121.56 (CH); 130.12 (CH); 132.19 (CH);137.67 (C); 138.26 (C); 148.26 (CH); 159.80 (C); 161.12 (C); 164.24 (C); 164.99 (C). Anal. Calcd for C24H20BrN3O5: C, 56.48; H, 3.95; N, 8.23; found: C, 56.34; H, 3.94; N, 8.21.

*3-Methoxyphenethyl 1-{2-[(4-bromophenyl)amino]-2-oxoethyl}-5-cyano-2-methyl-6-oxo-1,6-dihydropyridine-3-carboxylate* (**4e**), Yield = 11%; mp = 159–161 °C (EtOH); ^1^H-NMR (DMSO-d_6_) δ 2.66 (s, 3H, CH_3_); 2.98 (t, 2H, CH_2_*CH_2_*, *J* = 8.0 Hz) 3.71 (s, 3H, OCH_3_); 4.41 (t, 2H, OCH_2_, *J* = 8.0 Hz); 5.00 (s, 2H, N-CH_2_); 6.77 (d, 1H, Ar, *J* = 8.0 Hz); 6.88 (m, 2H, Ar); 7.20 (t, 1H, Ar, *J* = 8.0 Hz); 7.49 (m, 4H, Ar); 8.38 (s, 1H, Ar); 10.62 (exch br s, 1H, NH). ^13^C-NMR (DMSO-d_6_) δ 18.71 (CH_3_); 34.62 (CH_2_); 48.88 (CH_2_); 55.41 (CH_3_); 66.20 (CH_2_); 99.71 (C); 109.79 (C); 112.51 (CH); 114.96 (CH); 115.80 (C); 115.87 (C); 116.38 (CH); 121.54 (CH); 129.48 (CH); 132.19 (CH); 138.26 (C);140.08 (C); 148.15 (CH); 159.81 (C); 160.98 (C); 164.24 (C); 164.99 (C). Anal. Calcd for C_25_H_22_BrN_3_O_5_: C, 57.26; H, 4.23; N, 8.01; found: C, 57.12; H, 4.22; N, 8.03.

#### 3.2.2. General Procedure for **3a**, **3d,e**, and **5**

A mixture of compound **1** (0.84 mmol), 3.25 mL of SOCl_2_, and 0.05 mL of Et_3_N was refluxed under stirring for 8 h, and after cooling, the excess of SOCl_2_ was removed under vacuum. To obtain compound **3a**, the residue was dissolved in 3 mL of anhydrous propanol, the mixture was stirred at room temperature for 4 h, and the precipitate formed was recovered by vacuum filtration and purified by crystallization from ethanol. To obtain compounds **3d** and **3e**, the residue was dissolved in 3 mL of anhydrous THF, and then 0.42–0.84 mmol of the appropriate aryl-alcohol was added. The mixture was stirred at room temperature for 3–4 h, and the organic layer was evaporated under reduce pressure. The final compounds were purified by flash column chromatography using toluene/ethyl acetate/acetic acid 8:2:1 for compound **3d** and cyclohexane/ethyl acetate 1:3 for compound **3e** as eluent and, finally, crystallized from ethanol.

*Propyl 5-cyano-2-methyl-6-oxo-1,6-dihydropyridine-3-carboxylate* (**3a**)**,** Yield = 22%; mp = 190–192 °C dec. (EtOH); ^1^H-NMR (CDCl_3_) δ 1.01 (t, 3H, CH_2_*CH_3_*, *J* = 8.0 Hz); 1.77 (m, 2H, *CH_2_*CH_3_); 2.83 (s, 3H, CH_3_); 4.24 (t, 2H, OCH_2_, *J* = 6.0 Hz); 8.49 (s, 1H, Ar); 12.79 (exch br s, 1H, NH). Anal. Calcd for C_11_H_12_N_2_O_3_: C, 59.99; H, 5.49; N, 12.72; found: C, 59.85; H, 5.48; N, 12.69.

*3-Methoxybenzyl 5-cyano-2-methyl-6-oxo-1,6-dihydropyridine-3-carboxylate* (**3d**), Yield = 23%; mp = 215–217 °C dec. (EtOH); ^1^H-NMR (DMSO-d_6_) δ 2.58 (s, 3H, CH_3_); 3.73 (s, 3H, OCH_3_); 5.20 (s, 2H, OCH_2_); 6.89 (m, 1H, Ar); 7.00 (m, 2H, Ar); 7.28 (*t*, 1H, Ar, J = 8.0 Hz); 8.45 (s, 1H, Ar); 12.98 (exch br s, 1H, NH). Anal. Calcd for C_16_H_14_N_2_O_4_: C, 64.42; H, 4.73; N, 9.39; found: C, 64.59; H, 4.73; N, 9.37.

*3-Methoxyphenethyl 5-cyano-2-methyl-6-oxo-1,6-dihydropyridine-3-carboxylate* (**3e**), Yield = 44%; mp = 173–175 °C (EtOH); ^1^H-NMR (DMSO-d_6_) δ 2.50 (s, 3H, CH_3_); 2.94 (t, 2H, CH_2_*CH_2_*, *J* = 6.4 Hz) 3.70 (s, 3H, OCH_3_); 4.36 (t, 2H, OCH_2_, *J* = 8.0 Hz); 6.67 (d, 1H, Ar, *J* = 8.0 Hz); 6.88 (m, 2H, Ar); 7.20 (t, 1H, Ar, *J* = 8.0 Hz); 8.32 (s, 1H, Ar); 12.96 (exch br s, 1H, NH). Anal. Calcd for C_17_H_16_N_2_O_4_: C, 65.38; H, 5.16; N, 8.97; found: C, 65.23; H, 5.17; N, 8.95.

*5-Cyano-2-methyl-6-oxo-1,6-dihydropyridine-3-carboxamide* (**5**), A mixture of compound **1** (0.84 mmol) and 1,1′-carbonyldiimidazole (CDI) (0.91 mmol) in anhydrous THF (15 mL) was stirred at room temperature. After 2 h, 1.5 mL of 28% NH_4_OH was added and the mixture was stirred at room temperature for an additional 30 min. Then, it was acidified with 6N HCl and extracted with ethyl acetate (3 × 10 mL). After evaporation of the solvent, final compound was purified by flash column chromatography using dichloromethane/methanol 8.5:1.5 as eluent. Compound **5** was, finally, crystallized from ethanol. Yield = 19%; mp = 247–248 °C (EtOH); ^1^H-NMR (DMSO-d_6_) δ 2.46 (s, 3H, CH_3_); 7.37 (exch br s, 1H, NH); 7.71 (exch br s, 1H, NH); 8.23 (s, 1H, Ar); 12.72 (exch br s, 1H, NH). Anal. Calcd for C_8_H_7_N_3_O_2_: C, 54.24; H, 3.98; N, 23.72; found: C, 54.41; H, 3.97; N, 23.76.

*1-{2-[(4-Bromophenyl)amino]-2-oxoethyl}-5-cyano-2-methyl-6-oxo-1,6-dihydropyridine-3-carboxamide* (**6**), Compound **6** was obtained, starting from intermediate **5**, following the same procedure described for **2** and **4a**–**e**. The final compound was obtained by filtration in vacuo of the suspension and purified by flash chromatography using dichloromethane/methanol 9:1 as eluent and then crystallized from ethanol. Yield = 14%; mp = 283–284 °C dec. (EtOH); ^1^H-NMR (DMSO-d_6_) δ 2.47 (s, 3H, CH_3_); 4.96 (s, 2H, N-CH_2_); 7.49 (exch br s, 1H, NH); 7.52 (m, 4H, Ar); 7.91 (exch br s, 1H, NH); 8.22 (s, 1H, Ar); 10.63 (exch br s, 1H, NH). ^13^C-NMR (DMSO-d_6_) δ 18.67 (CH_3_); 48.72 (CH_2_); 99.88 (C); 115.69 (C); 116.14 (C); 116.47 (C); 121.49 (CH); 132.19 (CH); 138.34 (C); 147.32 (CH); 156.54 (C); 159.94 (C); 165.23 (C); 167.52 (C). Anal. Calcd for C_16_H_13_BrN_4_O_3_: C, 49.38; H, 3.37; N, 14.40; found: C, 49.52; H, 3.38; N, 14.37.

#### 3.2.3. General Procedure for **7a–e**

A mixture of compound **1** (0.84 mmol), 3.25 mL of SOCl_2_ and 0.05 mL of Et_3_N was refluxed under stirring for 8 h. After cooling, the excess of SOCl_2_ was removed under vacuum, the residue was dissolved in 4–5 mL of anhydrous THF and 0.84 mmol of appropriate alkylamine and 1.68 mmol of Et_3_N was added. The mixture was stirred at room temperature for 3–5 h and the organic layer was evaporated under reduce pressure. The final compounds were purified by flash column chromatography using dichloromethane/methanol 8.5:1.5 for compound **7a** and dichloromethane/methanol 9:1 for compounds **7b**–**e** as eluent. All final compounds were then crystallized from ethanol.

*5-Cyano-N,2-dimethyl-6-oxo-1,6-dihydropyridine-3-carboxamide* (**7a**), Yield = 21%; mp = 252–254 °C (EtOH); ^1^H-NMR (DMSO-d_6_) δ 2.42 (s, 3H, CH_3_); 2.67 (m, 3H, HN-*CH_3_*); 8.13 (s, 1H, Ar); 8.21 (exch br m, 1H, *NH-*CH_3_); 12.74 (exch br s, 1H, NH). Anal. Calcd for C_9_H_9_N_3_O_2_: C, 56.54; H, 4.75; N, 21.98; found: C, 56.38; H, 4.74; N, 21.93.

*5-Cyano-N-ethyl-2-methyl-6-oxo-1,6-dihydropyridine-3-carboxamide* (**7b**), Yield = 30%; mp = 247–250 °C (EtOH); ^1^H-NMR (DMSO-d_6_) δ 1.05 (t, 3H, CH_2_*CH_3_*, *J* = 8.0 Hz); 2.41 (s, 3H, CH_3_); 3.16 (m, 2H, NH*CH_2_*CH_3_); 8.17 (s, 1H, Ar); 8.20 (exch br m, 1H, *NH-*CH_2_); 12.72 (exch br s, 1H, NH). Anal. Calcd for C_10_H_11_N_3_O_2_: C, 58.53; H, 5.40; N, 20.48; found: C, 58.71; H, 5.41; N, 20.43.

*5-Cyano-2-methyl-6-oxo-N-propyl-1,6-dihydropyridine-3-carboxamide* (**7c**), Yield = 33%; mp = 244-246 °C (EtOH); ^1^H-NMR (DMSO-d_6_) δ 0.85 (t, 3H, CH_2_*CH_3_*, *J* = 8.0 Hz); 1.45 (m, 2H, *CH_2_*CH_3_); 2.41 (s, 3H, CH_3_); 3.09 (m, 2H, HN-*CH_2_*CH_2_); 8.16 (s, 1H, Ar); 8.20 (exch br m, 1H, *NH-*CH_2_); 12.72 (exch br s, 1H, NH). Anal. Calcd for C_11_H_13_N_3_O_2_: C, 60.26; H, 5.98; N, 19.17; found: C, 60.41; H, 5.99; N, 19.11.

*N-Butyl-5-cyano-2-methyl-6-oxo-1,6-dihydropyridine-3-carboxamide* (**7d**), Yield = 45%; mp = 252–254 °C (EtOH); ^1^H-NMR (DMSO-d_6_) δ 0.85 (t, 3H, CH_2_*CH_3_*, *J* = 8.0 Hz); 1.28 (m, 2H, *CH_2_*CH_3_); 1.42 (m, 2H, *CH_2_*CH_2_); 2.40 (s, 3H, CH_3_); 3.13 (t, 2H, HN-*CH_2_*CH_2_
*J* = 7.8 Hz); 8.16 (s, 1H, Ar); 8.20 (exch br m, 1H, *NH-*CH_2_); 12.73 (exch br s, 1H, NH). Anal. Calcd for C_12_H_15_N_3_O_2_: C, 61.79; H, 6.48; N, 18.01; found: C, 61.95; H, 6.47; N, 18.06.

*5-Cyano-N,N,2-trimethyl-6-oxo-1,6-dihydropyridine-3-carboxamide* (**7e**), Yield = 40%; mp = > 300 °C dec. (EtOH); ^1^H-NMR (DMSO-d_6_) δ 2.19 (s, 3H, CH_3_); 2.90 (s, 3H, N-CH_3_); 2.93 (s, 3H, N-CH_3_); 8.02 (s, 1H, Ar); 12.71 (exch br s, 1H, NH). Anal. Calcd for C_10_H_11_N_3_O_2_: C, 58.53; H, 5.40; N, 20.48; found: C, 58.39; H, 5.39; N, 20.51.

#### 3.2.4. General Procedure for **8a–e**

Compounds **8a**–**e** were obtained, starting from intermediates **7a–e**, following the same procedure described for **2** and **4a–e**. The final compounds were purified by flash chromatography using dichloromethane/methanol 9.5:0.5 for compound **8a** and dichloromethane/methanol 9:1 for compounds 8b, c and 8e as eluent. All final compounds were crystallized from ethanol.

*1-{2-[(4-Bromophenyl)amino]-2-oxoethyl}-5-cyano-N,2-dimethyl-6-oxo-1,6-dihydropyridine-3-carboxamide* (**8a**), Yield = 16%; mp = >300 °C dec. (EtOH); ^1^H-NMR (DMSO-d_6_) δ 2.49 (s, 3H, CH_3_); 2.70 (m, 3H, HN-*CH_3_*); 4.95 (s, 2H, CH_2_CO); 7.49 (m, 4H, Ar); 8.18 (s, 1H, Ar); 8.38 (exch br m, 1H, *NH-*CH_3_); 10.61 (exch br s, 1H, NH). ^13^C-NMR (DMSO-d_6_) δ 18.66 (CH_3_); 26.56 (CH_3_); 48.63 (CH_2_); 98.94 (C); 115.82 (C); 116.40 (C); 121.51 (CH); 132.18 (CH); 138.10 (C); 147.08 (CH); 156.10 (C); 159.99 (C); 165.10 (C); 166.16 (C). Anal. Calcd for C_17_H_15_BrN_4_O_3_: C, 50.64; H, 3.75; N, 13.89; found C, 50.49; H, 3.74; N, 13.92.

*1-{2-[(4-Bromophenyl)amino]-2-oxoethyl}-5-cyano-N-ethyl-2-methyl-6-oxo-1,6-dihydropyridine-3-carboxamide* (**8b**), Yield = 24%; mp = 290 °C dec. (EtOH); ^1^H-NMR (DMSO-d_6_) δ 1.05 (t, 3H, CH_2_*CH_3_*, *J* = 7.8 Hz); 2.49 (s, 3H, CH_3_); 3.19 (m, 2H, NH*CH_2_*CH_3_); 4.96 (s, 2H, CH_2_CO); 7.49 (m, 4H, Ar); 8.18 (s, 1H, Ar); 8.43 (exch br m, 1H, *NH-*CH_2_); 10.62 (exch br s, 1H, NH). ^13^C-NMR (DMSO-d_6_) δ 14.75 (CH_3_); 18.63 (CH_3_); 34.57 (CH_2_); 48.58 (CH_2_); 98.22 (C); 115.86 (C); 116.46 (C); 116.63 (C); 121.54 (CH); 132.17 (CH); 138.34 (C); 147.07 (CH); 155.89 (C); 159.97 (C); 165.10 (C); 165.61 (C). Anal. Calcd for C_18_H_17_BrN_4_O_3_: C, 51.81; H, 4.11; N, 13.43; found C, 51.96; H, 4.11; N, 13.40.

*1-{2-[(4-bromophenyl)amino]-2-oxoethyl}-5-cyano-2-methyl-6-oxo-N-propyl-1,6-dihydropyridine-3-carboxamide* (8c), Yield = 15%; mp = 274–276 °C dec. (EtOH); ^1^H-NMR (DMSO-d_6_) δ 0.86 (t, 3H, CH_2_*CH_3_*, *J* = 8.0 Hz); 1.47 (m, 2H, CH_2_*CH_2_*CH_3_); 2.48 (s, 3H, CH_3_); 3.12 (m, 2H, NH*CH_2_*CH_2_); 4.95 (s, 2H, CH_2_CO); 7.49 (m, 4H, Ar); 8.17 (s, 1H, Ar); 8.45 (exch br m, 1H, *NH-*CH_2_); 10.62 (exch br s, 1H, NH). ^13^C-NMR (DMSO-d_6_) δ 11.96 (CH_3_); 18.69 (CH_3_); 22.61 (CH_2_); 41.53 (CH_2_); 48.72 (CH_2_); 98.82 (C); 115.72 (C); 116.47 (C); 116.62 (C); 121.50 (CH); 132.17 (CH); 138.34 (C); 147.16 (CH); 156.03 (C); 159.97 (C); 165.23 (C); 165.61 (C). Anal. Calcd for C_19_H_19_BrN_4_O_3_: C, 52.91; H, 4.44; N, 12.99; found C, 52.97; H, 4.43; N, 13.02.

*1-{2-[(4-Bromophenyl)amino]-2-oxoethyl}-N-butyl-5-cyano-2-methyl-6-oxo-1,6-dihydropyridine-3-carboxamide* (**8d**), Yield = 30%; mp = 261–263 °C dec. (EtOH); ^1^H-NMR (DMSO-d_6_) δ 0.87 (t, 3H, CH_2_*CH_3_*, *J* = 8.0 Hz); 1.30 (m, 2H, *CH_2_*CH_3_); 1.45 (m, 2H, *CH_2_*CH_2_CH_3_); 2.48 (s, 3H, CH_3_); 3.17 (m, 2H, NH*CH_2_*CH_2_); 4.96 (s, 2H, CH_2_CO); 7.50 (m, 4H, Ar); 8.17 (s, 1H, Ar); 8.42 (exch br m, 1H, *NH-*CH_2_); 10.62 (exch br s, 1H, NH). ^13^C-NMR (DMSO-d_6_) δ 14.14 (CH_3_); 18.68 (CH_3_); 20.08 (CH_2_); 31.40 (CH_2_); 48.70 (CH_2_); 98.82 (C); 115.72 (C); 116.45 (C); 116.78 (C); 121.50 (CH); 132.17 (CH); 138.33 (C); 147.16 (CH); 156.03 (C); 159.97 (C); 165.23 (C); 165.61 (C). Anal. Calcd for C_20_H_21_BrN_4_O_3_: 53.94; H, 4.75; N, 12.58; found C, 53,79; H, 4.76; N, 12.55.

*1-{2-[(4-Bromophenyl)amino]-2-oxoethyl}-5-cyano-N,N,2-trimethyl-6-oxo-1,6-dihydropyridine-3-carboxamide* (**8e**), Yield = 26%; mp = 122–124 °C (EtOH); ^1^H-NMR (DMSO-d_6_) δ 2.30 (s, 3H, CH_3_); 2.85 (s, 3H, N-CH_3_); 2.94 (s, 3H, N-CH_3_); 4.94 (s, 2H, CH_2_CO); 7.50 (m, 4H, Ar); 8.12 (s, 1H, Ar); 10.62 (exch br s, 1H, NH). ^13^C-NMR (DMSO-d_6_) δ 18.78 (CH_3_); 34.98 (CH_3_); 39.55 (CH_3_); 48.68 (CH_2_); 100.05 (C); 115.73 (C); 115.92 (C); 116.38 (C); 121.50 (CH); 132.18 (CH); 138.32 (C); 146.20 (CH); 153.24 (C); 159.95 (C); 165.29 (C); 166.45 (C). Anal. Calcd for C_18_H_17_BrN_4_O_3_: C, 51.81; H, 4.11; N, 13.43; found C, 51.93; H, 4.10; N, 13.46.

*5-(4-methoxybenzoyl)-6-methyl-2-oxo-1,2-dihydropyridine-3-carboxamide* (**9**), A mixture of compound **1** (0.56 mmol) and 4 mL of Eaton’s reagent (P_2_O_5_ 7.7 wt % in MsOH) was heated at 60 °C under stirring for 20 min. Then, 0.56 mmol of anisole was added, and the mixture was stirred at 60 °C for additional 2 h. After cooling, cold water was added (10 mL), the precipitate was recovered by vacuum filtration, and the final compound was purified by crystallization from ethanol. Yield = 80%; mp = 273–275 °C (EtOH); ^1^H-NMR (DMSO-d_6_) δ 2.38 (s, 3H, CH_3_); 3.83 (s, 3H, OCH_3_); 7.05 (d, 2H, Ar, *J* = 8.0 Hz); 7.61 (exch br s, 1H, NH); 7.65 (d, 2H, Ar, *J* = 8.0 Hz); 8.22 (s, 1H, Ar); 8.78 (exch br s, 1H, NH); 12.87 (exch br s, 1H, NH). Anal. Calcd for C_15_H_14_N_2_O_4_: C, 62.93; H, 4.93; N, 9.79; found: C, 62.76; H, 4.94; N, 4.91.

*1-{2-[(4-bromophenyl)amino]-2-oxoethyl}-5-(4-methoxybenzoyl)-6-methyl-2-oxo-1,2-dihydropyridine-3-carboxamide* (**10**), Compound **10** was obtained, starting from compound **9**, following the same procedure described for **2** and **4a–e**. The final compound was purified by flash chromatography using dichloromethane/methanol 9:1 as eluent and finally crystallized from ethanol**.** Yield = 14%; mp = 269–272 °C (EtOH); ^1^H-NMR (DMSO-d_6_) δ 2.40 (s, 3H, CH_3_); 3.84 (s, 3H, OCH_3_); 5.02 (s, 2H, COCH_2_); 7.08 (d, 2H, Ar, *J* = 8.0 Hz); 7.53 (m, 4H, Ar); 7.69 (exch br s, 1H, NH); 7.72 (d, 2H, Ar, *J* = 8.0 Hz); 8.22 (s, 1H, Ar); 8.69 (exch br s, 1H, NH); 10.67 (exch br s, 1H, NH). ^13^C-NMR (DMSO-d_6_) δ 19.24 (CH_3_); 48.80 (CH_2_); 56.16 (CH_3_); 114.73 (CH); 115.66 (C); 116.63 (C); 118.33 (C); 121.52 (CH); 132.16 (CH); 132.58 (CH); 138.44 (C); 139.50 (C); 143.12 (CH); 155.08 (C); 162.01 (C); 164.07 (C); 164.31 (C); 165.59 (C); 192.97 (C). Anal. Calcd for C_23_H_20_N_3_O_5_: C, 55.44; H, 4.05; N, 8.43; found: C, 55.35; H, 4.04; N, 8.45.

*N-(4-bromophenyl)-2-[3-cyano-5-(4-methoxybenzoyl)-6-methyl-2-oxopyridin-1(2H)-yl]acetamide* (**11**), A solution of compound 10 (0.06 mmol) in 0.8 mL of POCl_3_ was refluxed under stirring for 1 h. After cooling, cold water was added (10 mL), and the precipitate was recovered by suction and purified by crystallization from ethanol. Yield = 34%; mp = 241–243 °C (EtOH); ^1^H-NMR (DMSO-d_6_) δ 2.34 (s, 3H, CH_3_); 3.85 (s, 3H, OCH_3_); 5.01 (s, 2H, COCH_2_); 7.08 (d, 2H, Ar, *J* = 8.0 Hz); 7.53 (m, 4H, Ar); 7.76 (d, 2H, Ar, *J* = 8.0 Hz); 8.20 (s, 1H, Ar); 10.64 (exch br s, 1H, NH). ^13^C-NMR (DMSO-d_6_) δ 19.55 (CH_3_); 48.78 (CH_2_); 56.20 (CH_3_); 99.41 (C); 114.80 (CH); 115.77 (C); 116.25 (C); 118.45 (C); 121.53 (CH); 129.94 (C); 132.20 (CH); 132.78 (CH); 138.30 (C); 147.47 (CH); 156.52 (C); 160.02 (C); 164.30 (C); 165.20 (C); 191.89 (C). Anal. Calcd for C_23_H_18_N_3_O_4_: C, 57.51; H, 3.78; N, 8.75; found: C, 57.36; H, 3.79; N, 8.73.

#### 3.2.5. General Procedure for **13b–c**

To a solution of compound **12** [24] (0.37 mmol) in anhydrous CH_3_CN (5 mL), 0.37 mmol of the appropriate 3- or 4-hydroxyphenyl-2-chloroacetamide **[29,30]** and 0.74 mmol of K_2_CO_3_ were added, and the mixture was refluxed for 5–6 h. After cooling, ice-cold water was added (20 mL), and the precipitate was recovered by suction: the final compounds were purified by column flash chromatography using cyclohexane/ethyl acetate 1:4 for compound **13a** and CH_2_Cl_2_/CH_3_OH 10:1 for **13b** and **13c** as eluent. Finally, all final compounds were crystallized from ethanol.

*2-[3-Cyano-5-(3-methoxyphenyl)-6-methyl-2-oxopyridin-1(2H)-yl]-N-(3-hydroxyphenyl) acetamide* (**13b**), Yield = 27%; mp = 128–130 °C (EtOH); ^1^H-NMR (DMSO-d_6_) δ 2.33 (s, 3H, 6-CCH_3_); 3.77 (s, 3H, OCH_3_); 4.98 (s, 2H, NCH_2_CO); 6.45 (d, 1H, Ar, *J* = 7.8 Hz); 6.87 (d, 2H, Ar, *J* = 8.4 Hz). 6.94 (d, 2H, Ar, *J* = 8.0 Hz); 7.05 (d, 1H, Ar, *J* = 8.4 Hz); 7.08 (s, 1H, Ar); 7.35 (t, 1H, Ar, *J* = 7.8 Hz); 8.12 (s, 1H, Ar); 9.40 (exch br s, 1H, OH); 10.33 (exch br s, 1H, NH). ^13^C-NMR (DMSO-d_6_) δ 19.52 (CH_3_); 49.07 (CH_2_); 55.66 (CH_3_); 99.63 (C); 106.75 (CH); 110.29 (CH); 111.23 (CH); 114.02 (CH); 115.57 (CH); 116.82 (C); 119.98 (C); 122.37 (CH); 129.97 (CH); 130.18 (CH); 138.89 (C); 140.05 (C); 149.09 (CH); 158.18 (C); 158.26 (C); 159.83 (C); 160.05 (C); 165.04 (C). Anal. Calcd for C_22_H_19_N_3_O_4_: C, 67.86; H, 4.92; N, 10.79; found: C, 68.05; H, 4.91; N, 10.76.

*2-[3-Cyano-5-(3-methoxyphenyl)-6-methyl-2-oxopyridin-1(2H)-yl]-N-(4-hydroxyphenyl) acetamide* (**13c**), Yield = 48%; mp = 127–129 °C (EtOH); ^1^H-NMR (DMSO-d_6_) δ 2.33 (s, 3H, 6-CCH_3_); 3.77 (s, 3H, OCH_3_); 4.96 (s, 2H, NCH_2_CO); 6.68 (d, 2H, Ar, *J* = 8.8 Hz); 6.86 (s, 1H, Ar). 6.88 (s, 1H, Ar); 6.95 (d, 1H, Ar, *J* = 8.0 Hz); 7.32-7.38 (m, 3H, Ar); 8.11 (s, 1H, Ar); 9.20 (exch br s, 1H, OH); 10.20 (exch br s, 1H, NH). ^13^C-NMR (DMSO-d_6_) δ 19.51 (CH_3_); 48.83 (CH_2_); 55.66 (CH_3_); 99.60 (C); 114.01 (CH); 115.56 (2xCH); 116.83 (C); 119.98 (C); 121.04 (2xCH); 122.10 (CH); 122.37 (CH); 130.18 (CH); 130.67 (C); 138.89 (C); 149.04 (CH); 153.46 (C); 154.04 (C); 159.82 (C); 160.06 (C); 164.40 (C). Anal. Calcd for C_22_H_19_N_3_O_4_: C, 67.86; H, 4.92; N, 10.79; found: C, 68.07; H, 4.93; N, 10.77.

#### 3.2.6. General Procedure for **14a–c**

To a cooled solution of the appropriate intermediate 1**3a–c** (**13a [24]**) (0.28 mmol) in anhydrous CH_2_Cl_2_ (5 mL), 0.56–1.12 mmol of a solution of 1M BBr_3_ in CH_2_Cl_2_ was added, and the mixture was stirred at room temperature for 2–6 h. After dilution with ice-cold water (20 mL) and neutralization with 30% NH_4_OH, the precipitate was recovered by suction and purified by column flash chromatography using as eluent toluene/ethyl acetate/acetic acid 8:2:1 for compound 1**4b**, and toluene/ethyl acetate/acetic acid 6:4:1 for compound 1**4c**. For compound 1**4a**, the suspension was extracted with CH_2_Cl_2_ (3 × 15 mL), and the organic layer was dried over Na_2_SO_4_ and evaporated in vacuo. The obtained residue was purified by column flash chromatography using CH_2_Cl_2_/CH_3_OH 10:1 as eluent. Then, compounds **14a–c** were crystallized from ethanol.

*N-(4-Bromophenyl)-2-[3-cyano-5-(3-hydroxyphenyl)-6-methyl-2-oxopyridin-1(2H)-yl] acetamide* (**14a**), Yield = 26%; mp = 282–284 °C dec. (EtOH); ^1^H-NMR (DMSO-d_6_) δ 2.32 (s, 3H, 6-CCH_3_); 5.00 (s, 2H, NCH_2_CO); 6.66 (s, 1H, Ar); 6.70 (d, 1H, Ar, *J* = 8.0 Hz). 6.77 (d, 1H, Ar, *J* = 8.4 Hz); 7.23 (t, 1H, Ar, *J* = 8.0 Hz); 7.47-7.54 (m, 4H, Ar); 8.08 (s, 1H, Ar); 9.62 (exch br s, 1H, NH); 10.33 (exch br s, 1H, OH). ^13^C-NMR (DMSO-d_6_) δ 19.36 (CH_3_); 50.56 (CH_2_); 101.48 (C); 113.86 (CH); 115.30 (CH); 115.48 (C); 117.07 (C); 120.98 (CH); 121.71 (2xCH); 122.19 (C); 130.09 (CH); 131.83 (2xCH); 136.72 (C); 137.87 (C); 148.48 (CH); 151.35 (C); 159.93 (C); 161.07 (C); 164.19 (C). Anal. Calcd for C_21_H_16_BrN_3_O_3_: C, 57.55; H, 3.68; N, 9.59; found: C, 57.73; H, 3.67; N, 9.56.

*2-[3-Cyano-5-(3-hydroxyphenyl)-6-methyl-2-oxopyridin-1(2H)-yl]-N-(3-hydroxyphenyl) acetamide* (14b), Yield = 39%; mp = 187–189 °C dec. (EtOH); ^1^H-NMR (DMSO-d_6_) δ 2.31 (s, 3H, 6-CCH_3_); 4.96 (s, 2H, NCH_2_CO); 6.68 (m, 4H, Ar); 6.77 (d, 1H, Ar, *J* = 8.0 Hz). 7.22 (t, 1H, Ar, *J* = 7.8 Hz); 7.33 (d, 2H, Ar, *J* = 8.4 Hz); 8.06 (s, 1H, Ar); 10.24 (exch br s, 1H, NH). ^13^C-NMR (DMSO-d_6_) δ 19.42 (CH_3_); 48.79 (CH_2_); 99.58 (C); 115.26 (CH); 115.65 (2xCH); 116.83 (C); 116.97 (CH); 120.24 (C); 120.44 (CH); 121.33 (2xCH); 130.21 (CH); 130.63 (C); 138.78 (C); 148.83 (CH); 153.28 (C); 154.10 (C); 158.03 (C); 160.08 (C); 164.44 (C). Anal. Calcd for C_21_H_17_N_3_O_4_: found C, 67.19; H, 4.56; N, 11.19; C, 67.05; H, 4.57; N, 11.21.

*2-[3-Cyano-5-(3-hydroxyphenyl)-6-methyl-2-oxopyridin-1(2H)-yl]-N-(4-hydroxyphenyl) acetamide* (**14c**), Yield = 32%; mp = 232–234 °C dec. (EtOH); ^1^H-NMR (DMSO-d_6_) δ 2.31 (s, 3H, 6-CCH_3_); 4.99 (s, 2H, NCH_2_CO); 6.45 (d, 1H, Ar, *J* = 8.0 Hz); 6.67 (s, 2H, Ar). 6.77 (d, 1H, Ar, *J* = 8.0 Hz); 6.95 (d, 1H, Ar, *J* = 8.4 Hz); 7.05 (m, 2H, Ar); 7.21 (t, 1H, Ar, *J* = 8.2 Hz); 8.06 (s, 1H, Ar); 10.46 (exch br s, 1H, NH). ^13^C-NMR (DMSO-d_6_) δ 19.43 (CH_3_); 49.01 (CH_2_); 99.58 (C); 106.79 (CH); 110.20 (CH); 111.27 (CH); 115.32 (CH); 116.82 (C); 117.03 (CH); 120.30 (C); 120.44 (CH); 129.93 (CH); 130.17 (CH); 138.73 (C); 140.04 (C); 148.88 (CH); 153.28 (C); 158.26 (C); 158.32 (C); 160.08 (C); 165.06 (C). Anal. Calcd for C_21_H_17_N_3_O_4_: found C, 67.19; H, 4.56; N, 11.19; C, 67.01; H, 4.56; N, 11.22.

### 3.3. Biological Assays

#### 3.3.1. Cell Culture

Human promyelocytic leukemia HL60 cells stably transfected with FPR1 (FPR1-HL60 cells) or FPR2 (FPR2-HL60 cells) (kind gift from Dr. Marie-Josephe Rabiet, Université Joseph Fourier, Grenoble, France) were cultured in RPMI 1640 medium supplemented with 10% heat-inactivated fetal calf serum, 10 mM HEPES, 100 μg/mL streptomycin, 100 U/mL penicillin, and G418 (1 mg/mL). G418 was removed in the last round of culture before assays were performed.

#### 3.3.2. Isolation of Human Neutrophils

Blood was collected from healthy donors in accordance with a protocol approved by the Institutional Review Board at Montana State University. Neutrophils were purified from the blood using dextran sedimentation, followed by Histopaque 1077 gradient separation and hypotonic lysis of red blood cells. Isolated neutrophils were washed twice and resuspended in HBSS^-^. Neutrophil preparations were routinely >95% pure, as determined by light microscopy, and >98% viable, as determined by trypan blue exclusion.

#### 3.3.3. Ca^2+^ Mobilization Assay

Changes in intracellular Ca^2+^ were measured with a FlexStation II scanning fluorometer (Molecular Devices, Sunnyvale, CA, USA). The cells, suspended in Hank’s balanced salt solution without Ca^2+^ and Mg^2+^ but with 10 mM HEPES (HBSS^−^), were loaded with 1.25 μg/mL Fluo-4 AM dye and incubated for 30 min in the dark at 37 °C. After dye loading, the cells were washed with HBSS^-^ containing 10 mM HEPES, resuspended in HBSS^+^ containing Ca^2+^, Mg^2+^, and 10 mM HEPES (HBSS^+^), and aliquoted into the wells of flat-bottom, half-area-well black microtiter plates (2 × 10^5^ cells/well). For the evaluation of direct agonist activity, compounds of interest were added from a source plate containing dilutions of test compounds in HBSS^+^, and changes in fluorescence were monitored (λ_ex_ = 485 nm, λ_em_ = 538 nm) every 5 s for 240 s at room temperature after the automated addition of compounds. Maximum change in fluorescence during the first 3 min, expressed in arbitrary units over baseline, was used to determine a response. Responses for FPR1 agonists were normalized to the response induced by 5 nM fMLF for FPR1-HL60 cells and neutrophils, or 5 nM WKYMVM for FPR2-HL60 cells, which were assigned a value of 100%. Curve fitting (5–6 points) and calculation of median effective inhibitory concentrations (IC_50_) were performed by nonlinear regression analysis of the dose–response curves generated using Prism 9 (GraphPad Software, Inc., San Diego, CA, USA).

#### 3.3.4. Animals

Male Sprague Dawley rats (Envigo, Varese, Italy) weighing approximately 200–250 g at the beginning of the experimental procedure were used. Animals were housed in Ce.S.A.L. (Centro Stabulazione Animali da Laboratorio, University of Florence) and used at least one week after their arrival. Four rats were housed per cage (size 26 × 41 cm^2^); animals were fed a standard laboratory diet, given tap water ad libitum, and kept at 23 ± 1 °C with a 12 h light/dark cycle, light at 7 a.m. All animal manipulations were carried out according to the Directive 2010/63/EU of the European parliament and of the European Union council (22 September 2010) on the protection of animals used for scientific purposes. The ethical policy of the University of Florence complies with the Guide for the Care and Use of Laboratory Animals of the US National Institutes of Health (NIH Publication No. 85–23, revised 1996; University of Florence assurance number: A5278-01). Formal approval to conduct the experiments described was obtained from the Italian Ministry of Health (No. 517/2017, 06/04/2017) and from the Animal Subjects Review Board of the University of Florence. Experiments involving animals have been reported according to ARRIVE guidelines. All efforts were made to minimize animal suffering and to reduce the number of animals used.

#### 3.3.5. CFA-Induced Inflammatory Arthritis

Articular damage was induced by injection of CFA (Sigma-Aldrich, Burlington, MA, USA) into the tibiotarsal joint [32,33]. Briefly, the rats were lightly anesthetized by 2% isoflurane. After sterilizing the left leg skin with 75% ethyl alcohol, the lateral malleolus was located by palpation and a 28-gauge needle was inserted vertically to penetrate the skin and turned distally for insertion into the articular cavity at the gap between the tibiofibular and tarsal bone until a distinct loss of resistance was felt. A volume of 50 μL of CFA was then injected (left paw, ipsilateral). Control rats received 50 μL of saline solution in the tibiotarsal joint. The paw pressure and the incapacitance tests (see below) were performed 8 days after CFA injection.

#### 3.3.6. Drug Administration

**AMC3** (10 mg/kg) was suspended in 1% carboxymethyl cellulose and administered p.o. daily for 7 days starting from the day of CFA injection. Behavioral measurements were conducted 24 h after the last treatment (day 8).

#### 3.3.7. Paw Pressure Test

Nociceptive threshold was determined with an analgesimeter (Ugo Basile, Varese, Italy). Briefly, a constantly increasing pressure was applied to a small area of the dorsal surface of the hind paw using a blunt conical mechanical probe. Mechanical pressure was increased until vocalization or a withdrawal reflex occurred while rats were lightly restrained. Vocalization or withdrawal reflex thresholds were expressed in grams. An arbitrary cut-off value of 100 g was adopted. The data were collected by an observer who was blinded to the protocol [34].

#### 3.3.8. Incapacitance Test

Weight bearing changes were measured using an Incapacitance Apparatus (Linton Instrumentation, UK), detecting changes in postural equilibrium after a hind limb injury. As described by Di Cesare Mannelli et al. [35], rats were trained to stand on their hind paws in a box with an inclined plane (65° from horizontal). This box was placed above the incapacitance apparatus. This allowed us to independently measure the weight that the animal applied on each hind limb. The value reported for each animal is the mean of five consecutive measurements. In the absence of hind limb injury, rats applied an equal weight on both hind limbs, indicating postural equilibrium, whereas an unequal distribution of weight on the hind limbs indicated a monolateral decreased pain threshold. Data are expressed as the difference between the weight applied to the limb contralateral to the injury and the weight applied to the ipsilateral one (Δ weight).

#### 3.3.9. Statistical Analysis

Each value represents the mean ± SEM of five rats per group. Groups were as shown in the figures, and different groups were used for acute treatments. The analysis of variance was performed by an ANOVA. A Bonferroni’s significant difference procedure was used as a post hoc comparison. P values of less than 0.05 or 0.01 were considered significant. Data were analyzed using the ‘Origin 8.1′ software.

### 3.4. Molecular Modelling

Structures of compounds **AMC4**, **14a**, and **14c** were built using ChemOffice 2016 software, pre-optimized with the MM2 force field and saved in Tripos MOL2 format. A homology model of FPR1 with docked *f*MLF peptide [31] and a cryo-EM structure of FPR2-G_i_ complex with the peptide agonist Trp-Lys-Tyr-Met-Val-D-Met-NH_2_ [31] (PDB entry 6OMM) were taken as sources of the receptor geometries for the docking study. Each of the receptor structures was then imported into the Molegro Virtual Docker 6.0 program (MVD) together with the built models of ligands **AMC4**, **14a**, and **14c**. A search space for docking was defined as a sphere 12 Å in radius located at the geometric center of gravity of the bound peptide molecule (*f*MLF for FPR1 or WKYMVM for FPR2). MolDock score functions were applied with a 0.3 Å grid resolution. Flexibility of ligands was accounted for with respect to torsions auto-detected in MVD. The receptor structures were considered rigid. The “Internal HBond” and “sp^2^-sp^2^ torsions” options were activated in the “Ligand evaluation” panel of the MVD Docking Wizard. Three hundred docking runs were performed for each investigated compound with each receptor. The option “Return multiple poses for each run” was enabled, and the post-processing options “Energy minimization” and “Optimize H-bonds” were applied after docking. Similar poses were clustered at an RMSD threshold of 1 Å.

## 4. Conclusions

In this paper, we report new pyridinone derivatives as analogues of the potent FPR agonists **AMC3** and **AMC4**, previously described by us. These compounds were differently elaborated at position 5 of pyridinone scaffold and at the phenyl acetamide chain The biological results indicated that position 5 of pyridinone tolerated different substituents, while elimination of bromine in the para-position of the phenyl acetamido group and/or the insertion of an OH group resulted in compounds that were completely devoid of activity. Most of the products acted as mixed FPR1/FPR2 agonists with activity in the submicromolar/micromolar range and exhibited moderate selectivity toward FPR2.

These results were confirmed by docking studies, which showed that replacing the bromine atom with a hydroxy group drastically changed the ligand binding mode and mutually alternated positions of the N-phenylacetamido and phenylpyridinone moieties in both the FPR1 and FPR2 binding sites.

Finally, the activity of **AMC3** in a rat model of RA at 10 mg/kg/day for one week, in agreement with its previously published acute pain-relieving efficacy [24], makes this compound interesting for further pharmacological investigation.

## Data Availability

Not available.

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
