# Peer review of "Pyridinone Derivatives as Interesting Formyl Peptide Receptor (FPR) Agonists for the Treatment of Rheumatoid Arthritis"

_molecules, 2021, doi:10.3390/molecules26216583_

Round 1

Reviewer 1 Report

Reviewer comments

This manuscript describes “Pyridinone Derivatives as Interesting Formyl Peptide Recep-tors (FPRs) Agonists for the Treatment of Rheumatoid Arthritis”. This is interesting work on N-formyl peptide receptors as anti-RA compounds. This manuscript is well written and very well tested compounds in various assays. This is useful work and can be consider for publication. Still, there are few shortcomings that will preclude its publication in the current form.

Major and minor concerns:

  1. In 2.1. Chemistry section, authors need to cite all relevant references related to all synthetic steps.
  2. In scheme 1, 2 and 4 and tables, authors need to use proper format of italics for i & n
  3. Molecular docking, authors need to list important interactions in a table so readers can understand it easily.
  4. Docking figure titles needs to be more informative and clearer. Need to report key interactions as well.
  5. % Purify of final compounds needs to report in manuscript. As authors claimed that they used TLC and NMR for purity, but these methods are not reliable for final purity of compounds.

Reviewer 2 Report

This manuscript represents a continuation of this group efforts towards the design and synthesis of pyridinone derivatives as FPR Agonists for the treatment of rheumatoid arthritis.

The simulations of active vs inactive compounds based on docking is fair. I think that the qualitative docking calculations results are exaggerated and a more complete picture can be obtained using MD simulations. I will not ask MD simulations from the authors but to restrict the docking calculations discussion at least moving the 2/3 of the relevant discussion in the Supporting Information. Only few clues are needed and this extended docking calculations part is consistent with overestimation, since more accurate binding free energy calculations can be performed to investigate the SARs.

Reviewer 3 Report

The manuscript entitled “Pyridinone derivatives as interesting formyl peptide receptors (FPRs) agonists for the treatment of rheumatoid arthritis” by Giovannoni et al. is devoted to develop a new series of pyridinone derivatives as potential candidates for the treatment of rheumatoid arthritis. The paper is a SAR extension of a recent study reported by the same group (Bioorg. Chem., 2020, 100, 103880). There is no doubt regarding the usefulness of this paper for SAR in this class of derivatives with potential as FPRs agonists, even if the degree of novelty is not very high and the biological results are not spectacular for the new compounds.

I recommend authors to fix the following issues:

- the authors should put into only one table, the data from tables 1-3. All three types of structures can be placed in the first row of the table, and data would be easier to follow.

-several general conclusions would be useful for the readers at the end of the manuscript.

- in the experimental section: for most compounds, it appears that melting points were determined for derivatives crystallized from EtOH, but only for few of them this is clear from the experimental procedure they were crystallized from ethanol. Please check if it is a copy-paste mistake, or otherwise clarify this aspect in the experimental procedure.

-also, there a few typos to be revised in the manuscript.

Round 2

Reviewer 2 Report

The authors have adequately addressed my comments as regards the biomolecular modeling and I suggest the publication of this interesting manuscript.